# Natural Stabilizers and Nanostructured Lipid Carrier Entrapment for Photosensitive Compounds, Curcumin and Capsaicin

**DOI:** 10.3390/pharmaceutics16030412

**Published:** 2024-03-17

**Authors:** Wipanan Jandang, Chadarat Ampasavate, Kanokwan Kiattisin

**Affiliations:** 1Department of Pharmaceutical Sciences, Faculty of Pharmacy, Chiang Mai University, Chiang Mai 50200, Thailand; wipanan_jandang@cmu.ac.th; 2Center for Excellent in Pharmaceutical Nanotechnology, Faculty of Pharmacy, Chiang Mai University, Chiang Mai 50200, Thailand

**Keywords:** chili extract, turmeric extract, capsaicin, curcumin, photostability, nanostructured lipid carriers

## Abstract

Capsaicin and curcumin, the active components of chili and turmeric, are prone to instability when exposed to light. Therefore, this research aimed to enhance the photostability of both extracts via the use of antioxidants, natural sunscreen, and nanostructured lipid carriers (NLCs). NLCs were chosen for this this study due to their advantages in terms of stability, drug loading capacity, occlusive effect, skin penetration, and controlled release. The photostability of each extract and extracts mixed with antioxidants, including grape seed extract, tea extract, and chlorogenic acid, were determined. Chlorogenic acid can enhance the photostability of capsaicin from 6.79 h to 16.50 h, while the photostability of curcumin increased from 9.63 h to 19.25 h. In addition, the use of natural sunscreen (sunflower oil) also increased the photostability of capsaicin and curcumin. The mixed extracts were then loaded into NLCs. The particle size of the formulation was 153.73 nm with a PDI value of 0.25. It exhibited high entrapment efficiency (more than 95%). In addition, it effectively reduced the decomposition of capsaicin and curcumin. Importantly, the natural stabilizers chosen for NLC fabrication significantly improved the photostability of curcumin and capsaicin by 600% and 567% compared to the unstabilized counterparts. This improvement contributes to the sustainability and bioavailability of these compounds in both cosmeceutical and pharmaceutical products.

## 1. Introduction

Combining natural ingredients in herbal remedies has been widely accepted and used in foods, food supplements, pharmaceutics, and cosmetics. Knowledge of these combinations is particularly significant in the development of antioxidant and anti-inflammatory products, as this knowledge can enable the improvement of both their activities and stability [1,2]. Chili and turmeric have been demonstrated to possess high antioxidant, anti-aging, anti-obesity, anti-cancer, and anti-cellulite properties [3,4]. Chili and turmeric extracts represent an effective treatment of several skin conditions. The main component in turmeric and chili extracts are curcumin and capsaicin, respectively [3,5]. Curcumin has been used to treat psoriasis, atopic dermatitis, iatrogenic dermatitis, wounds, skin cancer, skin infection, herpes, and cellulitis [6,7]. Capsaicin has been used as an anti-inflammatory compound in psoriasis, pruritus, allergic dermatitis, as well as in extended pain syndrome [8]. However, capsaicin becomes unstable when exposed to light [9], and curcumin can be degraded by light and alkaline solution [10,11]. This instability constitutes is a major limitation, particularly when including these compounds in pharmaceutical and cosmetic fields. In recent years, several studies have documented the photodegradation of curcumin; both artificial and natural light can destroy the structure of curcumin. Moreover, curcumin rapidly degrades in a citrate–phosphate buffer at pH 3 (k 5.842 × 10^−^^5^ min^−^^1^). In an aqueous buffer at pH 8.0, it exhibits a constant rate of 280 × 10^−^^3^ h^−^^1^, corresponding to one half-life (T_1/2_) of 2.5 h. Therefore, curcumin in turmeric can degrade in both light and an alkaline solution [12].

Increasing the stability of photosensitive compounds can be achieved using various methods such as adding antioxidants or sunscreens, developing nano-delivery systems, and using sun protection packaging for storage. Related research showed that photostability can be increased by adding natural antioxidants and natural sunscreens. Generally, natural antioxidants are used in cosmetic or nutraceutical products such as coffee, grape seed, and tea extracts [2,13,14]. Previous research reported the increasing levels of photostability of plant extracts with phenolic compounds by via copigmentation [15]. In addition, sunscreen or a sun protection agent can be added to prevent the light-induced degradation of the extracts. It helps to reflect or absorb light or to scavenge free radicals stimulated by light; this can ensure that the extract is protected from natural sunlight or artificial light [9,12]. Sunscreen can be divided into three types: physical, chemical, and natural sunscreen. Each sunscreen offers distinct mechanisms of protection against the harmful effects of ultraviolet (UV) radiation. Physical sunscreens contain mineral compounds such as zinc oxide or titanium dioxide as their active ingredients. These work via the formation of a physical barrier on the skin, reflecting or scattering UV radiation away from the skin’s surface. Chemical sunscreens contain organic (carbon-based) compounds such as avobenzone, octocrylene, or oxybenzone are largely the active ingredients. They absorb UV radiation, transforming it into heat, which is then released from the skin. Natural sunscreens are natural extracts from plants or natural oils that have antioxidant activity, anti-inflammatory activity, soothing effects, or low sun protection value (SPF) to protect against UV radiation [16]. Natural sunscreens were tested under accelerated photodegradation conditions according to ICH guidelines [17]. The result showed that reducing photodegradation can be beneficial. Research by Oresajo and colleagues (2010) compared products with antioxidants and products with antioxidants combined with sunscreen among volunteers. The results showed that antioxidants combined with sunscreen provided greater effectiveness in sun protection and presented higher antioxidant activity than formulas containing antioxidants alone [18]. When natural sunscreens were combined with antioxidants, they could protect the active substances in extracts against sunlight [19]. Moreover, nano-delivery systems can also obstruct the degradation of extracts. Extracts are trapped within the particles, making them difficult to decompose. The fabrication of topical dosage forms containing nanostructured lipid carriers (NLCs) loaded with stable curcumin and capsaicin as active compounds was pursued in this study. These are suitable for the containment of oil-soluble extracts. In addition, NLCs with a small particle size can deliver active ingredients directly to the skin’s lipid layer. In addition, they help to reduce the side effects of active compounds, increase skin penetration, increase stability, and generate continuous release [20]. When comparing NLCs with solid lipid nanoparticles (SLNs), NLCs have demonstrated clear advantages in terms of skin penetration, stability, and drug loading capacity than SLNs [21]. This superiority can be attributed to the unique composition of NLCs, which includes a combination of both solid and liquid lipids. Our strategies have emphasized the protection of the photosensitive compounds via the selection of natural protective agents such as antioxidants from plant extracts and oils demonstrating antioxidant or sun protection activities.

Therefore, this research is interested in studying the effects of natural antioxidants, including grape seed extract, tea extract, and chlorogenic acid, on the stability of capsaicin and curcumin. In addition, natural sunscreens, including sunflower oil, olive oil, and coconut oil, were used to prevent the photodegradation of the blended chili and turmeric extracts. After that, the extract mixture was loaded into NLCs to increase their stability and specific properties for topical applications. The results from this research can be further used to improve the photostability of chili and turmeric extracts for topical cosmeceutical or pharmaceutical products. Extract-loaded NLCs can be used as anti-cellulite products for cosmetic application. Moreover, these can also be used as anti-inflammation products for the treatment of skin diseases like psoriasis or atopic dermatitis.

## 2. Materials and Methods

### 2.1. Materials

Chili extract and capsaicin (63.04%) were received from Bangkok Lab and Cosmetic (Bangkok, Thailand). The tea extract was purchased from Specialty Natural Product (Bangkok, Thailand). Grape seed extract, sunflower oil, olive oil, coconut oil, glyceryl distearate, cetyl alcohol, homosalate, isopropyl myristate, and glyceryl behenate were obtained from Chanjao Longevity (Bangkok, Thailand). The turmeric extract was acquired from Welltech Biotechnology (Bangkok, Thailand). Caffeine (99.4%), chlorogenic acid (95%), curcumin (98%), trolox, gallic acid, 2,2′-diphenyl-1-picrylhydrazyl (DPPH), ferric tripyridyltriazine (TPTZ), linoleic acid, and nitro blue tetrazolium chloride (NBT) were procured from Sigma-Aldrich (St. Louis, MO, USA). Standard chlorogenic acid (96.6%) was purchased from BOC Sciences natural products (Shirley, NY, USA). Nicotinamide adenine dinucleotide (NADH) and phenazine methosulfate (PMS) were purchased from TCI (Bangkok, Thailand). Resveratrol (99.94%) was obtained from MedChemExpress (Bangkok, Thailand). Water, ethanol, methanol, acetonitrile, hydrochloric acid, di-sodium hydrogen phosphate dihydrate (Na_2_HPO_4_·2H_2_O), sodium dihydrogen phosphate dihydrate (NaH_2_PO_4_·2H_2_O), and iron (II) chloride (FeCl_2_) were acquired from RCI Labscan (Bangkok, Thailand). Phosphotungstic acid was procured from Ted Pella, Inc. (Redding, CA, USA). Furthermore, 2,2′-Azobis(2-methylpropionamidine) dihydrochloride (AAPH) was purchased from ACROS Organics (Bangkok, Thailand). Ammonium thiocyanate (NH_4_SCN) was obtained from Kemaus (Bangkok, Thailand). Ortho-phosphoric acid was acquired from Merck Millipore (Berlin, Germany).

### 2.2. Methods

#### 2.2.1. Chemical Composition Analysis Using High Performance Liquid Chromatography (HPLC)

Chili extract, turmeric extract, tea extract, grape seed extract, and chlorogenic acid were analyzed to determine the amount of outstanding chemical markers in each sample using the Agilent 1100 Series HPLC Value System and 1100 detector (VWD) (Santa Clara, CA, USA). The chemical markers of chili extract, turmeric extract, tea extract, and grape seed extract are capsaicin, curcumin, caffeine, and resveratrol, respectively. In addition, chlorogenic acid, an active compound found in coffee extracts, demonstrated stabilization properties in our previous findings [9]. In all the extracts, chlorogenic acid and chemical markers in were prepared with methanol. The reverse phase chromatography gradient elution was used following the conditions shown in Table 1. All marker compounds can be analyzed using the same HPLC condition with slight modifications from Boonrueang, N. (2019) [9,22,23,24,25]. Chromatographic separation was achieved with a Purospher^®^ STAR C-18 analytical column (RP-18 end-capped, 150 mm × 4.6 mm) with a gradient elution of 0.1% *v*/*v* phosphoric acid in water, acetonitrile, and methanol at the initial ratio of 83.5:8.25:8.25 for 7 min, followed by a gradual change to the ratio of 40:55:5 for 30 min, and then a return to the initial ratio of 83.5:8.25:8.25 for 2 min, as shown in Table 1. The flow rate was 1 mL/min. The samples were detected using dual wavelengths at 280 nm (capsaicin), 425 nm (curcumin), 330 nm (chlorogenic acid), 280 nm (caffeine), and 306 nm (resveratrol), as shown in Table 2. This method was used to calculate the percentage of chemical markers in each extract, photostability test, percent entrapment efficiency (%EE), and percent loading capacity (%LC) of extract-loaded NLCs.

#### 2.2.2. Photostability Test

The photostability of the single extract and the extract with antioxidants were both evaluated following the experiment of Boonrueang, N. (2019) [9], in accordance with the photostability testing protocol outlined in the ICH Q1B guidelines [17]. The natural antioxidant agents in this study have sun protection properties. From previous investigations, proanthocyanidins and resveratrol in grape seed extracts and epigallocatechin-3-gallate (EGCG) in tea extracts exhibited the potential to increase the sun protection factor (SPF) values of other extracts when prepared in the extract mixture [26,27]. Chlorogenic acid possesses good photochemical stability, and does not degrade under UVA or UVB irradiation [14]; therefore, this was chosen to be used in combination with chili and turmeric extracts. Mixtures were commonly prepared by the mol:mol ratio of active markers. Single extracts, double mixtures, and triple mixtures were prepared using quantities of chemical marker compounds, calculated to the same concentrations of 100 µM. The selected antioxidants in this study were chosen from the reported substances with high antioxidant properties, and can be mixed with turmeric and chili extracts.

The ratios of the marker compounds in the extract mixture were modified from previous studies [28,29]. The molar ratios of capsaicin/curcumin/chlorogenic acid, capsaicin/curcumin/resveratrol, and capsaicin/curcumin/caffeine in the extract mixture were 1:1:1, 1:1:2, 2:1:1, 2:1:2, and 2:1:4.

The extract mixtures according to the ratios mentioned above were prepared in methanol. The sample mixtures and the single chili and turmeric extracts in methanol were exposed to UV and white light at 6000 to 7000 lux in the light chamber for 0, 4, and 8 h, with adjustments made for the duration [9]. At each time point, the samples were collected and analyzed for the remaining amounts of the marker compound using HPLC. The sample volume was adjusted to 2 mL before HPLC analysis. The log decrease in the extract concentration versus the time change were used to calculate the rate constant (K) as described below: [9]
(1)$$-Kt=I\eta \frac{\left[A\right]t}{\left[A\right]0}$$
where [A]_t_ is the concentration at each time, [A]_0_ is the concentration at time 0, and K is the first-order rate constant.

Curcumin and capsaicin were degraded using first-order kinetics [30,31]. Therefore, a rate constant was used to calculate the half-life (T_1/2_) of capsaicin and curcumin, as shown below:(2)$$T_{1/2}=\frac{0.693}{K}$$

#### 2.2.3. Anti-Oxidation Activity of Extract Mixtures with Antioxidative Agents

The extract mixture with natural antioxidative agents with the highest presented photostability from the photostability test in Photo Stability Section was selected to study antioxidant activity as it compared with that of a single extract.

##### DPPH (2,2-diphenyl-1-picrylhydrazyl) Radical Scavenging Assay

The single extract and the extract mixture with the selected antioxidant agent were both evaluated for radical scavenging activity using the DPPH assay. The DPPH solution was prepared at a concentration of 0.033 mg/mL. Altogether, 20 µL of sample solution was added in a 96-well plate and mixed with 180 µL of DPPH solution. Absorbance levels were measured at a wavelength of 517 nm after 30 min of incubation using a microplate reader (Green Lane, NY, USA) [32]. The positive control comprised Trolox. The percentage of inhibition was calculated using the following equation:(3)$$\%Inhibition=\frac{\left(A0-A1\right)}{A0}\times 100$$
where A_0_ is the absorbance of control and A_1_ is the absorbance of the tested sample.

The half maximal inhibitory concentration (IC_50_) was calculated from the linear equation of the graph plotted between %inhibition and concentrations. Then, the y value was substituted with 50% to obtain the x value, which is IC_50_.

##### Ferric Reducing Antioxidant Power (FRAP) Assay

The single extract and the extract mixture with the selected antioxidant agent’s ferric-reducing ability were evaluated using the FRAP assay. The FRAP solution consisted of 300 mM of TPTZ 0.0156 g, 5 mL of hydrochloric acid, 0.0270 g of ferric (III) chloride in 5 mL of water, and an acetate buffer (50 mL, 300 mM, pH 3.6). The sample solution was added to a 96-well plate and mixed with the FRAP reagent. Then, the sample was incubated at room temperature for 4 min, and the absorbance was read at 593 nm using microplate readers (Green Lane, NY, USA). The positive control comprised Trolox. The FRAP value was calculated from a standard curve of ferrous sulfate and then reported as M FeSO_4_/g extract [33].

##### Lipid Peroxidation Inhibition Using the Ferric Thiocyanate Assay

The single extract and the extract mixture with the selected antioxidant agent’s lipid peroxidation inhibition activity were evaluated using the ferric thiocyanate assay. In total, 2 µL of the sample was mixed with 350 µL of linoleic acid and 50 µL of AAPH, and was then incubated in a water bath at 50 °C for 4 h. Then, 2 µL of iron (II) chloride (FeCl_2_) and 2 µL of ammonium thiocyanate (NH_4_SCN) were added in a 96-well plate and incubated in the dark at 37 °C for 3 min. The absorbance was measured at 500 nm using a microplate reader (Green Lane, NY, USA) [34], and the positive control comprised Trolox. The percentage of inhibition was calculated using the Equation (3) and calculated IC_50_.

##### Superoxide Radical Scavenging Activity Assay

The single extract and the extract mixture with the selected antioxidant agent’s superoxide radical scavenging activity were evaluated using the superoxide radical scavenging activity assay. In all, 20 µL of the sample was mixed with 50 µL of 129 µM NBT, 50 µL of 498 µM NADH, and 50 µL of 81 µM PMS in a 96-well plate. Then, the sample was incubated in the dark at room temperature for 15 min. The sample was measured for absorbance at a wavelength of 540 nm using microplate readers (Green Lane, NY, USA) [33]. The positive control comprised Trolox. The percentage of inhibition was calculated using the Equation (3) and calculated IC_50_.

#### 2.2.4. Compatibility of Chili Extract, Turmeric Extract, and Chlorogenic Acid Determined Using Differential Scanning Calorimetry (DSC)

The chili extract, turmeric extract, chlorogenic acid, and chili extract mixed with turmeric extract and chlorogenic acid were studied in terms of their thermal behavior using DSC 25 TA Instruments (New Castle, DE, USA). Samples were scanned from 20 to 200 °C at a scanning rate of 10 °C/min. An empty aluminum pan was used as a reference under a nitrogen atmosphere with a flushing rate of 40 mL/min [35].

#### 2.2.5. Natural Oil Selection for the Photostability Test and the Development of Nanostructured Lipid Carriers

##### Extract Solubility

Each extract (10 mg) was dissolved in sunflower oil, coconut oil, and olive oil. Then, the solubility was assessed according to the British Pharmacopoeia [36,37].

##### Antioxidant Activity of Oils Using DPPH Radical Scavenging Assay

The radical scavenging activity of sunflower oil, coconut oil, and olive oil were evaluated using the DPPH assay. The method was tested using the method in DPPH (2,2-diphenyl-1-picrylhydrazyl) Radical Scavenging Assay Section. The results were presented as the percentage inhibition, and the positive control comprised Trolox.

##### SPF Value of Oils

The initial stock solution was prepared via taking 1% *v*/*v* of each oil in an ethanol/water solution in the ratio of 40:60. Thereafter, the absorbance values of each sample were determined from 290 to 320 nm [38]. The SPF value was calculated using the equation below:(4)$$SPF=CF\times \sum_{290}^{320}EE(\lambda )\times I\left(\lambda \right)Abs(\lambda )$$
where CF is the correction factor (10), EE (λ) is the erythemogenic effect of radiation with wavelength λ, and Abs (λ) are the spectrophotometric absorbance values at wavelengths from 290 to 320.

##### Photostability of Mixed Chili and Turmeric Extracts with Selected Antioxidants and Natural Oils

The mixed chili and turmeric extracts with the best antioxidants were prepared and mixed with olive oil, coconut oil, and sunflower oil in methanol. Then, the samples were evaluated for photostability using the same method in Photostability Test Section.

#### 2.2.6. Development of NLCs

NLCs were prepared via a high-energy method using a high shear homogenizer at 20,000 rpm for 15 min. The major components of NLCs comprise solid lipid (glyceryl distearate), liquid lipids (isopropyl myristate, sunflower oil, and olive oil), emulsifiers (polysorbate 80 and sorbitan oleate), and distilled water [39,40]. The compositions of NLCs were varied to determine the most appropriate formulation (small droplet size, well-dispersed particles, and good stability), as shown in Table 3 [41].

##### Characterization and Stability of NLCs

The particle size, zeta potential, and PDI of NLCs were measured using the Zetasizer (London, UK). In addition, the stability of NLCs was evaluated using a centrifuge and was then stored at room temperature for one month. The physical appearance, particle size, zeta potential, and PDI were evaluated before and after the stability test.

##### Development of Extract-Loaded NLCs

The best NLCs (small particle size, PDI less than 0.3 and good stability) were selected to load the extract mixture. The extract-loaded NLCs were prepared using the same method in Development of NLCs Section. The extract mixture used to load in NLCs consisted of 0.04 g of chili extract, 0.02 g of turmeric extract and 0.08 g of chlorogenic acid in sunflower oil.

##### Entrapment Efficiency (EE) and Loading Capacity (LC) of Extract-Loaded NLCs

EE and LC were determined using an indirect method. Briefly, extract-loaded NLCs 500 µL were mixed with 500 µL of methanol in an Eppendorf tube (total compound added) and centrifuged at 7000 rpm for 10 min at room temperature. The extract-loaded NLCs 500 µL were added to an Amicon^®^ ultracentrifuge tube and then centrifuged at 3000 rpm for 60 min at room temperature (free nonentrapped compound) then the sample was analyzed using HPLC of the %EE and LC were calculated using the equation below: [42].
(5)$$\%EE=\frac{(Total\;compound\;added-Free\;compound)\times 100}{Total\;compound\;added}$$
(6)$$\%LC=\frac{Weight\;of\;capsaicin/curcumin\;in\;lipid\;nanoparticles\times 100}{Weight\;of\;lipid\;nanoparticles}$$

##### Morphology of Extract-Loaded NLCs Using Transmission Electron Microscopy (TEM)

The morphology of extract-loaded NLCs was evaluated using TEM (a JEOL JEM-2100 Electron Microscope, Tokyo, Japan). The sample was diluted with water in a ratio of 1:10 before being dropped onto a 300 mesh copper grid and air dried for 15 min. Then, 1% *w*/*v* phosphotungstic acid, soluble in 30% ethanol, was dropped, and the excess chemical was removed using a Millipore Filter 0.22 µm. The grid was analyzed using TEM operated at 100 kV at 25,000× magnification [42].

##### Characterization and Photostability of Extract-Loaded NLCs

The formulations were measured for particle size, zeta potential, and PDI using the Zetasizer (London, UK). The amount of capsaicin and curcumin in the formulation was analyzed using HPLC. In addition, the stability of the formulation in a stability vial was evaluated at room temperature for one month (25 °C), including a photostability test via exposure to UV and white light. After the stability test, the NLC sample (1 mL) was mixed with 4 mL of methanol in a centrifuge tube. The sample was then centrifuged, and the supernatant was collected for analysis using HPLC.

#### 2.2.7. Statistical Analysis

Statistical analysis was calculated by a Paired sample *t*-test and One-way ANOVA using the SPSS Program version 17.0.

## 3. Results and Discussion

### 3.1. Percentages of Chemical Marker Compounds in Extracts by HPLC

From the research of Boonrueang, N. (2019) [10], a gradient was developed via the adjustment of the time and the use of phosphoric acid instead of acetic acid. This condition can separate and analyze every substance. Table 4 and Appendix A show the percentages of the chemical markers in the extracts and an analytical grade substance. The retention times of capsaicin in the analytical grade substance and the chili extract were found to be 10.01 min and 10.82 min. The percentage of the capsaicin in the chili extract determined using HPLC was 4.26%. The retention times of curcumin in an analytical grade substance and in the turmeric extract were detected at 24.49 min and 24.74 min, respectively. The percentage of curcumin in the turmeric extract was 69.7%. The retention times of chlorogenic acid in an analytical grade substance and in chlorogenic acid standard were found to be 7.41 min and 7.40 min, respectively. The percentage of chlorogenic acid was 75%. The retention time of resveratrol in an analytical grade substance and in the grape seed extract was found to be 15.98 min and 15.84 min, respectively. The percentage of resveratrol in the grape seed extract was 85%. The retention time of caffeine in an analytical grade substance and the tea extract was found to be 9.00 min and 9.05 min, respectively. The percentage of caffeine in the tea extract was 4.64%. From the experiment, the amounts of chemical markers obtained from comparing the corresponding standards were less than those specified in the COAs. This might have been due to the decomposition of substances from storage and containers [5,7], emphasizing that re-analysis is highly recommended in order to obtain an exact amount of the active markers.

### 3.2. Photostability of Single Extract and the Extracts with Antioxidants

The chili extract containing capsaicin exhibited photodegradation with half-lives of 6.79 ± 0.18 h upon exposure to UV and white light. The turmeric extract containing curcumin demonstrated half-lives of 9.53 ± 1.51 h. When the chili extract was mixed with the turmeric extract at a ratio of 2:1, the half-life of capsaicin was significantly decreased when compared with a single extract (*p* < 0.05). When the chili and turmeric extracts were mixed with the grape seed extract, the tea extract, or chlorogenic acid, the photodegradation of the chili extract significantly decreased (*p* < 0.05). On the other hand, the photodegradation of the turmeric extract when mixed with all antioxidants did not significantly differ from a single extract (*p* < 0.05). The half-life of the extract mixture when adding chlorogenic acid experienced a longer decay time than the grape seed extract and tea extract. All results are shown in Table 5. The information of capsaicin in terms of the decomposition under light exposure and stabilization strategy is rather scarce. The chili extract was reported to increase the photostability of the turmeric extract. A change was observed when mixing the two substances. The half-life of the turmeric extract increased from 14.14 h to 19.56 h, which was similar to the results found in this research [9]. Other researchers studied the effect of chlorogenic acid and resveratrol in terms of the degradation of naproxen under UV irradiation in a black-light cabinet for 24 h. The results showed that the remaining amount of naproxen when mixed with resveratrol following UV irradiation exposure was 99.4 ± 1.1%, while the remaining amount of naproxen when mixed with chlorogenic acid showed a moderate photoprotection effect of 94.1 ± 0.7%. The remaining amounts of naproxen mixed with chlorogenic acid and resveratrol were significantly higher when compared with those of naproxen alone (84.1 ± 3.8%) [43]. Optimal compounds and ratios were proven to mitigate the degradation of natural photosensitive compounds such as curcumin and capsaicin in this study. The degradation of curcumin has been widely accepted through the autoxidation pathway, resulting in heptadienedione chain cleavage being reported by Wang et.al. (1997) [44]. The autoxidation degradation of curcumin in a buffer solution is spontaneous, initiated by O_2_-accepted free radicals from curcumin before rapidly propagating to all curcumin molecules in the solution [40]. Nimiya et.al. (2016) [45] reported the co-addition of diverse redox active antioxidants (gallic acid, ascorbate, vitamin C, TBHQ, caffeic acid, rosmarinic acid, and Trolox) significantly increased the stability of the curcumin. The stabilization was proven by the protection of free radicals, which occurred at the phenolic functional group instead of the hydroxyl ion groups in the center of the curcumin molecules [45]. Unlike curcumin, the photostability data of capsaicin is very limited. It was reported that a low concentration (0.5 µM) of capsaicin was unstable when compared to a higher concentration (4 µM), whether protected from light or not. Fifty percent of capsaicin kept at room temperature was lost after 4 months [46]. The molecular structure of capsaicin is similar to the half structure of curcumin; therefore, interactions at 4-hydroxy-3-methoxyphenyl moiety of both compounds were primarily stabilized with the redox active antioxidants. When chlorogenic acid was added, it may interact with the phenolic moiety that subsequently reduced the free radical generation of curcumin and capsaicin [47,48].

### 3.3. Anti-Oxidation Activity of the Extract Mixture with Antioxidant Agents

From the photostability test of the chili extract and turmeric extract that were mixed with chlorogenic acid at a ratio of 2:1:4, they were further studied for their antioxidant activity. The antioxidant activity of the chili extract, turmeric extract, extract mixture (chili mixed with turmeric and chlorogenic acid), and standards using different assays is presented in Figure 1. Using the DPPH method, the turmeric extract showed anti-radical activity equal to that of the chili extract and the extract mixture (*p* > 0.05). However, a single extract and the extract mixture showed less activity than Trolox (*p* < 0.05). From the lipid peroxidation inhibition assay, the chili extract possessed good lipid peroxidation inhibition which was comparable with Trolox (*p* > 0.05). In addition, the chili extract showed higher activity than the turmeric extract and the extract mixture (*p* < 0.05). In contrast, the inhibition activities of the turmeric extract and the extract mixture did not significantly differ (*p* > 0.05). The results from the superoxide radicals scavenging assay found that the extract mixture indicated superoxide radicals scavenging activity equal to that of the chili extract and gallic acid (*p* > 0.05). In addition, the extract mixture and the chili extract possessed the ability to inhibit superoxide radicals at a higher rate than the turmeric extract (*p* < 0.05). From the FRAP assay, the single extract and the extract mixture exhibited metal reducing property, equal to that of Trolox (*p* > 0.05). As with all results mentioned above, it could be observed that the chili extract possessed antioxidant activity via the lipid peroxidation inhibition, superoxide radical scavenging, and FRAP assays, while the extract mixture indicated antioxidant activity via the superoxide radical scavenging assay.

The in vitro evaluation of the antioxidant activity represents different mechanisms of anti-oxidation. The DPPH method constitutes an antiradical assay based on the electron transfer to DPPH radicals. When the number of electron pairs is complete, the inhibition of free radicals occurs [30]. The FRAP method is used to measure the ability of reducing agents in the redox-linked colorimetric method, where the ferric tripyridyltriazine complex is reduced by a substance with antioxidant activity. Antioxidant agents can donate electrons to Fe^3+^ and change the structure to Fe^2+^ [49]. The superoxide radical scavenging method is a chain reaction with increased free radicals within living cells. Therefore, excess free radicals must be removed. Hydrogen peroxide supplies oxygen to free radicals, and the inhibition of free radicals can occur. The last method, lipid peroxidation inhibition, is used to inhibit the decomposition of lipids from the oxidation of linoleic acid in order to stop inhibit the chain reaction [31,32].

### 3.4. DSC Analysis

DSC was used to investigate the thermal behavior of materials which were composed using either the single extract and the extract mixture. The DSC thermograms of the chili extract, turmeric extract, chlorogenic acid, and extract mixture are shown in Figure 2. The DSC thermograms of the chili extract (CH), the turmeric extract (TM), and chlorogenic acid (CA) were detected at 58.38, 175.63, and 86.14 °C, respectively. When the CH, TM, and CA were physically mixed at a ratio of 1:1:1, the endothermic peaks representing the melting of CH, TM, and CA were found at 56.91, 150.91, and 190.90 °C, respectively. A clear CH peak mix was found at a ratio of 1:1:1, while the other peaks were not noted in the DSC chromatogram. In addition, the extract mixture revealed no other peaks or new peaks, indicating that the substances possessed good compatibility. The calculation of the thermal behavior is shown in Table 6.

Related research studied the compatibility between oleic acid and the turmeric extract using DSC, which reported a result that was consistent with ours. The turmeric extract exhibited a sharp endothermic peak at 178.02 °C, which is consistent with its melting point. A slight change was observed in the melting point of the turmeric extract (170.23 °C) when oleic acid was added; however, the extract was still considered to possess good compatibility [50].

### 3.5. Selection of Natural Oils for Photostability Test and Development of NLCs

Solubility extracts and chlorogenic acid were evaluated using natural oils. The results are presented according to the British Pharmacopoeia, 2009 [36,37] in Appendix A. The chili extract, the turmeric extract, chlorogenic acid, and the grape seed extract are freely soluble in sunflower oil. In addition, the turmeric and chili extracts are freely soluble in olive oil, whereas the grape seed extract is freely soluble in both olive and coconut oils. In contrast, the tea extract cannot be dissolved in all oils as it dissolves well in water.

The antioxidant activity of oils was tested using the DPPH assay, as shown in Table 7. The results showed that sunflower oil presented the highest inhibition value, followed by olive oil, while coconut oil did not exhibit any antioxidant activity. Sunflower, olive, and coconut oil were composed of oleic acid at 83.6, 79.7, and 2.58%, respectively [50,51,52,53]. Related research indicated that sunflower oil contained a high amount of oleic acid and long-chain carbons. This oil can provide more electrons to free radicals than olive and coconut oil [50]. Therefore, sunflower oil demonstrated the highest inhibiting value.

The SPF values are shown in Table 7. The results showed that sunflower oil possessed the highest SPF value, followed by olive and coconut oil, respectively. However, all natural oils have SPF values equal to homosalate with a nonsignificant difference (*p* > 0.05). Homosalate is a reference chemical sunscreen that absorbs UV rays in the UVA (320 to 400 nm) and UVB (290 to 320 nm) ranges. Related research indicates that natural sunscreens can absorb UVB (290 to 320 nm). In addition, sunflower oil can protect the skin from UVB [38].

### 3.6. Photostability of Mixed Chili and Turmeric Extracts with Selected Antioxidants and Natural Oil

Sunflower oil and olive oil were selected for use as natural sunscreens for mixing with the extract mixture. The chili extract, the turmeric extract, and chlorogenic acid in the ratio of 2:1:4 was dissolved in sunflower oil or olive oil, and evaluated for photostability. All half-lives were shown in Table 5. The results showed that the half-lives of capsaicin and curcumin dissolved in sunflower oil gave higher half-lives than olive oil. The half-life of capsaicin is 23.10 ± 0.25 h, and that of curcumin is 23.10 ± 0.94 h. Previous research reported that natural sunscreen helped to reduce the photodegradation of the chili extract and the turmeric extract better than when using antioxidants alone [18,19]. This is because the combination of two or more substances causes natural sunscreens to be more effective [12]. Additionally, the SPF value of natural sunscreens can protect the extract from sunlight.

### 3.7. Characterization and Stability of Unloaded NLCs

Glyceryl distearate was used as solid lipid in unloaded NLCs with a fixed concentration (3% *w*/*w*). Isopropyl myristate, sunflower oil, and olive oil were used as liquid lipids with concentrations in the range of 0.5–2% *w*/*w*. Polysorbate 80 and sorbitan oleate were used as emulsifiers at 20% *w*/*w*. The particle size, PDI, and zeta potential of all formulations are presented in Table 8. All formulas showed small particle sizes in the range of 98–192 nm, with PDI values ranging from 0.2 to 0.4. The PDI value of nanomaterials can be described as well dispersed (PDI range: 0.1–0.3), moderately dispersed (PDI range: 0.3–0.5), and aggregated (PDI range: 0.5–1) [54]. Formula 5 demonstrated the smallest droplet size (98.68 ± 0.32 nm). A formula using synthetic liquid lipids (isopropyl myristate) alone or a formula using sunflower and olive oils generated bigger particle sizes than a formula using isopropyl myristate with sunflower oil. Glyceryl distearate contains stearic acid with a carboxyl functional group similar to that of sunflower and olive oils. But isopropyl myristate has ester functional groups, causing it to assemble into a chemical structure, which is dissimilar to natural oils. When solid lipid is mixed with natural oil alone, it becomes saturated and generates a smaller particle size. Moreover, when solid lipid is mixed with isopropyl myristate and natural oil, the particle size is smaller than when using isopropyl myristate alone [55,56]. All formulas had a high zeta potential of more than −52 mV. Carboxyl and ester functional groups in the structure of the composition can generate high and negative zeta potential values. In conclusion, Formulas (2), (4), and (5) were selected to study their stability due to the desired particle size, an acceptable PDI, and a high zeta potential.

Formulas (2), (4), and (5) were selected to be studied in terms of their stability at room temperature for 30 days following centrifugation. The results are shown in Table 9, Table 10 and Table 11. The particle sizes of Formula (2) and Formula (4) were significantly changed following centrifugation (*p* < 0.05) when compared with day 0. Interestingly, the particle size of Formula 5 did not change following the stability test. The PDI values of all formulas did not change after the stability test. The zeta potential of all formulas increased after 30 days (*p* < 0.05), while this did not change after centrifugation. Therefore, Formula (5) was chosen to load the extract mixture.

### 3.8. Characterization and Stability of Extract-Loaded NLCs

The size, PDI, and zeta potential of extract-loaded NLCs before and after the stability test are presented in Appendix A. The results demonstrated that the particle size and PDI of extract-loaded NLCs were 153.73 ± 1.97 nm and 0.25 ± 0.00. The particle size and PDI of the formulation did not change following storage for 30 days (*p* > 0.05). The zeta potential of the formulation following the stability test was significantly different when compared to the initial value (*p* < 0.05). However, the zeta potential was shown to be in a good range between +30 mV to −30 mV. Zeta potential is a value that indicates the stability of the preparation. High negative values indicate a better formulation stability. The charge surrounding each particle is negatively charged, exerting a weak repulsive force between the particles to reduce coalescence. However, previous research reported that the zeta potential of the coffee extract loaded into the NLCs was increased during storage, but still remained stable [57].

Appendix A shows the percent %EE of capsaicin in extract-loaded NLCs at 95.94 ± 0.05 and the %EE of curcumin in extract-loaded NLCs at 97.79 ± 0.36. The results related to the research of Lee, H. J. (2020) [58], who used glyceryl distearate to entrap the turmeric extract (%EE in the range of 81.1–99.4%). From the results, we can infer that glyceryl distearate showed a high ability to entrap the extract. When trying to study the use of olive oil together with Geleol, it was found that olive oil was able to increase the EE with a %EE of 82.49% [59]. The %LC capacity of capsaicin in extract-loaded NLCs was 0.56 ± 0.00, and the %LC of curcumin in extract-loaded NLCs was 0.29 ± 0.00. The %LC was low because the amount of capsaicin in the formulation was 0.04 g. However, high capsaicin loading, up to 8.53%, was observed in the previous chili-loaded NLC formulation [35]. The use of capsaicin in topical products is limited to the range of 0.025–0.075 g for appropriate anti-inflammatory effects, leading to less skin irritation [60].

The morphology of the unloaded NLCs and the extract-loaded NLCs are shown in Figure 3. The obtained nanoparticles were spherical in shape and homogeneously colored, with the particle size distributed approximately between 98 and 153 nm.

### 3.9. Photostability of Extract-Loaded NLCs

The photostability of extract-loaded NLCs was evaluated after 30 days. The results are presented in Table 12. It was found that the half-lives of capsaicin in the chili extract and curcumin in the turmeric extract did not change following the photostability test.

Figure 4 showed the half-life of the extract alone, the extract mixture, and the extract mixture in sunflower oil compared with the extract-loaded NLC. The results showed that capsaicin in the chili extract initially decomposed after 6.79 ± 0.18 h. When the chili extract was mixed with the turmeric extract and chlorogenic acid, the stability of the chili extract increased, and the half-life was increased to 16.50 ± 1.11 h (*p* < 0.05). The half-life was also increased to 23.10 ± 0.25 h after adding the natural oil (*p* < 0.05). In addition, the development of extract-loaded NLCs can also help to protect capsaicin degradation, and the half-life was increased to 38.50 ± 0.14 h (*p* < 0.05). The curcumin in the turmeric extract initially decomposed under light exposure in 9.63 ± 1.51 h. When the chili extract and chlorogenic acid were added, the stability of the turmeric extract increased to 19.25 ± 1.23 h, and the half-life was also increased to 23.10 ± 0.94 h after adding the natural oil (*p* < 0.05). In addition, the development of extract-loaded NLCs can also help to protect curcumin degradation; the half-life was increased to 57.75 ± 1.96 h (*p* < 0.05). When the chili and turmeric extracts were combined with stabilizers, including antioxidant agents (chlorogenic acid) and natural oil or natural sunscreen (sunflower oil), this can increase the half-lives of each active compound within the extracts. When chlorogenic acid is mixed with chili and turmeric extracts, a chemical reaction occurs in the phenolic functional group, increasing the half-life of curcumin and capsaicin. Additionally, sunflower oil increases the half-lives of curcumin and capsaicin in the extract when exposed to light. This is because sunflower oil acts as an antioxidant and a chemical sunscreen, absorbing radiation and thus preventing skin damage. The results related to previous research where it was inferred that sunscreen can increase the stability of extracts better than using antioxidants alone [18]. When adding natural sunscreen presenting antioxidant activity, the extract mixture showed a synergistic effect, inhibiting the photodegradation of the marker compound by absorbing ultraviolet radiation [19]. Moreover, the NLCs can protect the light-induced degradation of the extract. Interestingly, capsaicin in the chili extract possessed a 567% increase in half-life, and curcumin in the turmeric extract possessed a 600% increase. The NLCs composed of solid lipids and liquid lipids provided space for the storing and protecting of active substances. Natural liquid lipids with a high SPF value and solid lipids indicate occlusive effects and photoprotective properties. In appropriate NLC formulations, the mixture of solid and liquid lipids becomes an amorphous structure, preventing polymorphic transitions of actives and solid lipids, thereby averting crystallization into a more ordered structure. This facilitates higher drug loading into the lipid matrix, preventing the expulsion of actives from the loaded lipid nanoparticles, enhancing their stability [61,62]. NLCs can be used in topical products for cosmetic or pharmaceutics applications due to a number of advantages, such as improved drug stability, occlusive effects, increasing skin penetration, controlling drug release, reducing side effects, etc. [61]. Previous research reported that curcumin-loaded NLCs demonstrated extended release, which proves beneficial for psoriasis treatment. The small particle size (less than 200 nm) of the NLC formulation exhibits a greater surface area, beneficial for skin permeation [63,64]. The 200 nm formulations presented smooth surfaces, as well as a good integrity of the lipid skin bilayer when compared to other particle sizes [65]. In addition, the occlusive effect of the NLC demonstrated benefits for skin moisture and skin penetration, particularly for thick and scaly skin, often found in patients with psoriasis [65,66]. Other studies have shown that the incorporation of sunscreens into NLCs can increase the SPF value of the sunscreen [66]. Therefore, it has potential to protect the active compounds from photodegradation.

## 4. Conclusions

The photostability of the chili extract, the turmeric extract, and the mixed extracts was determined using HPLC. The chili and turmeric extracts, when combined with the natural antioxidant and chlorogenic acid, demonstrated an increased photostability. In addition, the mixture of the extracts showed good compatibility. When the chili extract, the turmeric extract, and chlorogenic acid were mixed and dissolved in sunflower oil, the half-lives of the chili and turmeric extracts were increased. The extract-loaded NLCs demonstrated a small particle size, PDI values less than 0.3, a high zeta potential, a high percentage of entrapment efficiency, and a good loading capacity. The NLCs can increase the half-life of curcumin from 9.63 h to 57.75 h, and that of capsaicin from 6.79 h to 38.50 h. Interestingly, capsaicin in the chili extract possessed a 567% increase in half-life, and curcumin in the turmeric extract possessed a 600% increase in half-life. Therefore, capsaicin in the chili extract and curcumin in the turmeric extract can be stabilized by the addition of natural antioxidants, natural sunscreen, as well as entrapment in the NLCs. In addition, the extract-loaded NLCs can be further incorporated into topical cosmeceutical or pharmaceutical preparations that can generate many benefits, including good photostability, occlusive effects, controlled release, and penetration to the target site in the skin. Due to the biological activities of curcumin and capsaicin, they can also be used for anti-cellulite products in cosmetic applications. Moreover, they can be used as pharmaceutical products with anti-inflammatory qualities for skin diseases such as psoriasis and atopic dermatitis.

## Figures and Tables

**Figure 1 pharmaceutics-16-00412-f001:**
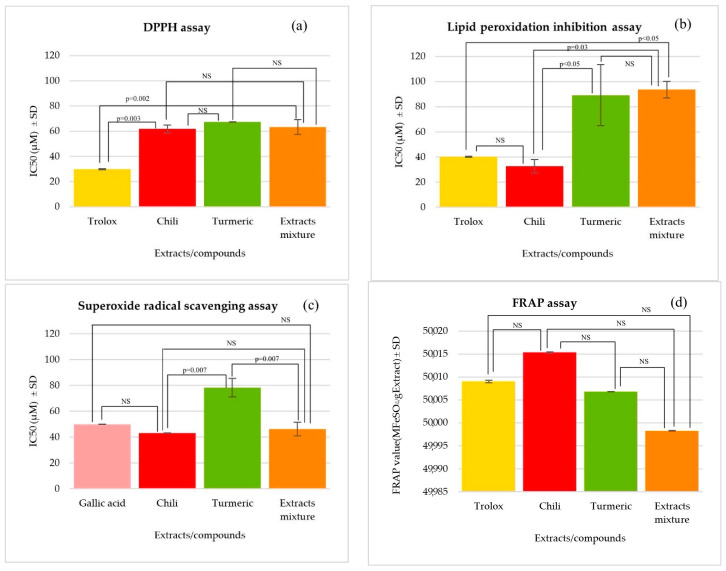
Antioxidant activity of the chili extract, turmeric extract, extract mixture, and standards evaluated using the (**a**) DPPH assay, (**b**) lipid peroxidation inhibition assay, (**c**) superoxide radical scavenging assay, and (**d**) FRAP assay. All tests were performed in triplicate. The *p* value above the bar graph shows a significant difference between samples at *p* < 0.05, and NS indicates a nonsignificant difference which was calculated using One-way ANOVA from the SPSS Program.

**Figure 2 pharmaceutics-16-00412-f002:**
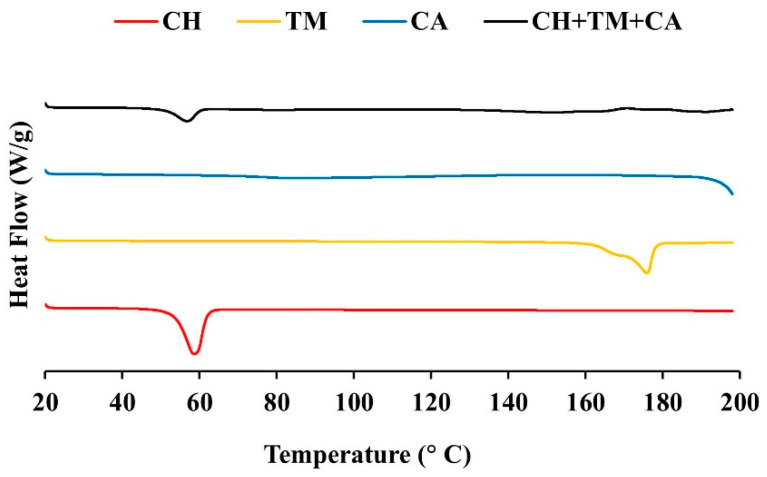
DSC thermograms of the chili extract (CH), the turmeric extract (TM), chlorogenic acid (CA), and the chili extract mixed with the turmeric extract and chlorogenic acid (CH + TM + CA).

**Figure 3 pharmaceutics-16-00412-f003:**
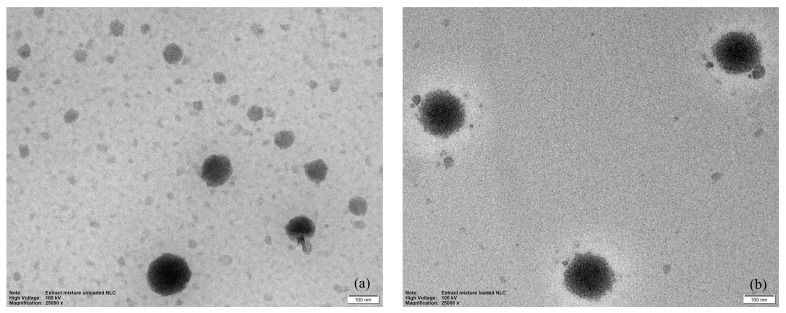
Transmission electron microphotographic image of unloaded NLCs (**a**) and extract-loaded NLCs (**b**).

**Figure 4 pharmaceutics-16-00412-f004:**
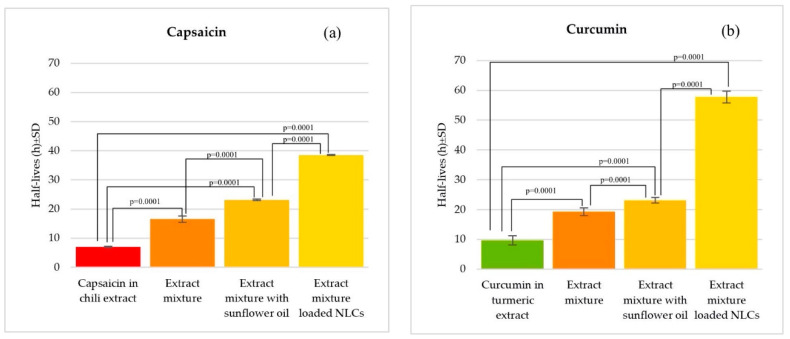
Comparison of the half-lives of capsaicin in the chili extract, the extract mixture, the extract mixture with sunflower oil, and the extract-loaded NLCs (**a**); the half-lives of curcumin in the turmeric extract, the extract mixture, the extract mixture with sunflower oil, and the extract-loaded NLCs (**b**) following 8 h of light exposure. The *p* value below 0.05 above the bar graph indicates a significant difference between each pair of samples, calculated by One-way ANOVA using the SPSS program.

**Table 1 pharmaceutics-16-00412-t001:** Gradient condition of HPLC.

Time (min)	Mobile Phase Line A 0.1% Phosphoric Acid in Water	Mobile Phase Line B Acetonitrile	Mobile Phase Line C Methanol
0	83.50	8.25	8.25
7	83.50	8.25	8.25
12	40.00	55.00	5.00
30	40.00	55.00	5.00
31	83.50	8.25	8.25
32	83.50	8.25	8.25

**Table 2 pharmaceutics-16-00412-t002:** Wavelength of extract/compound used in HPLC.

Extract/Compound	Chemical Marker	Wavelength (nm)
Chili	Capsaicin	280
Turmeric	Curcumin	425
Chlorogenic acid	Chlorogenic acid	330
Grape seed	Resveratrol	306
Tea	Caffeine	280

**Table 3 pharmaceutics-16-00412-t003:** Composition of NLCs.

Ingredient	HLB	% *w*/*w*					
		1	2	3	4	5	6
Glyceryl distearate	3.50	3.00	3.00	3.00	3.00	3.00	3.00
Isopropyl myristate	11.5	2.00	1.00	1.50	-	-	-
Sunflower oil	7.00	-	-	-	2.00	1.00	1.50
Olive oil	7.00	-	1.00	0.50	-	1.00	0.50
Polysorbate 80	15.00	4.49	2.80	3.64	1.12	1.12	1.12
Sorbitan oleate	4.30	15.51	17.20	16.36	18.88	18.88	18.88
Water (distilled)	-	75.00	75.00	75.00	75.00	75.00	75.00

**Table 4 pharmaceutics-16-00412-t004:** Percentage of the chemical marker in the extracts and analytical grade substances.

Extract/Compound	Chemical Marker	% Major Compound Specified in COA	% Chemical Marker from HPLC
Chili	Capsaicin	13.67	4.26
Turmeric	Curcumin	95.00	69.70
Chlorogenic acid	Chlorogenic acid	95.00	75.00
Grape seed	Resveratrol	98.00	85.00
Tea	Caffeine	13.70	4.64

**Table 5 pharmaceutics-16-00412-t005:** Half-lives of capsaicin and curcumin with antioxidants following the photostability test.

Extract/Compound	Ratio	Half-Lives (h)	
		Capsaicin	Curcumin
Chili	-	6.79 ± 0.18 ^a^	ND
Turmeric	-	ND	9.53 ± 1.51 ^a^
Chili/Turmeric	2:1	14.44 ± 0.79 ^b^	16.50 ± 0.67 ^a^
Chili/Turmeric/Chlorogenic acid (dissolved in methanol)	2:1:4	16.50 ± 1.11 ^c^	19.25 ± 1.23 ^a^
Chili/Turmeric/Chlorogenic acid (dissolved in sunflower oil)	2:1:4	23.10 ± 0.25 ^d^	23.10 ± 0.94 ^a^
Chili/Turmeric/Chlorogenic acid (dissolved in olive oil)	2:1:4	19.25 ± 0.24 ^c^	23.10 ± −0.18 ^a^

ND means not determined. Different letters in each column mean significant differences at *p* < 0.05 calculated using One-way ANOVA from the SPSS Program.

**Table 6 pharmaceutics-16-00412-t006:** Thermal behavior of the extract mixture.

Result	Single Compound			CH + TM + CA		
	CH	TM	CA	CH	TM	CA
Onset (°C)	54.01	168.94	67.40	51.41	133.83	182.01
Peak (°C)	58.38	175.63	86.14	56.91	150.91	190.90
Enthalpy (J/g)	114.28	106.52	52.24	34.26	12.97	9.58

Abbreviations: CH = Chili extract, TM = Turmeric extract, CA = Chlorogenic acid.

**Table 7 pharmaceutics-16-00412-t007:** Antioxidant activity of natural oils using the DPPH assay and SPF values of natural oils and standard sunscreen.

Natural Oil and Sunscreen	% Inhibition ± SD	SPF Value ± SD
Sunflower oil	90.59 ± 0.83 ^a^	19.72 ± 0.01 ^a^
Olive oil	80.12 ± 1.46 ^a^	10.19 ± 0.03 ^a^
Coconut oil	0 ^b^	19.51 ± 0.04 ^a^
Homosalate	NT	28.55 ± 0.02 ^a^

NT means not tested. Different letters mean significant differences at *p* < 0.05 calculated using One-way ANOVA from the SPSS Program.

**Table 8 pharmaceutics-16-00412-t008:** Characterizations of unloaded NLCs.

Formulations	Mean ± SD		
	Size (nm)	PDI	Zeta Potential (mV)
(1)	182.20 ± 0.79 ^b^	0.30 ± 0.03 ^a^	−52.50 ± 0.46 ^a^
(2)	105.63 ± 0.55 ^b^	0.27 ± 0.01 ^a^	−54.90 ± 0.26 ^a^
(3)	125.34 ± 1.45 ^b^	0.41 ± 0.00 ^b^	−52.43 ± 0.64 ^a^
(4)	117.83 ± 0.49 ^b^	0.26 ± 0.01 ^a^	−53.27 ± 2.56 ^a^
(5)	98.68 ± 0.32 ^a^	0.27 ± 0.00 ^a^	−53.57 ± 1.24 ^a^
(6)	192.87 ± 2.58 ^b^	0.41 ± 0.02 ^b^	−54.00 ± 0.50 ^a^

Different letters in each column are significantly different at *p* < 0.05, calculated by One-way ANOVA from the SPSS program.

**Table 9 pharmaceutics-16-00412-t009:** Particle size of unloaded NLCs before and after the stability test.

Formulations	Mean ± SD		
	Day 0	Day 30	Centrifuge
(2)	105.63 ± 0.55	107.53 ± 3.10	141.40 ± 2.56 *
(4)	117.83 ± 0.49	118.90 ± 2.86	102.83 ± 0.59 *
(5)	98.68 ± 0.32	98.25 ± 0.86	98.66 ± 0.52

Asterisk (*) is significantly different at *p* < 0.05 when day 0 and day 30 are compared; centrifuge calculated by a Paired sample *t*-test using the SPSS program.

**Table 10 pharmaceutics-16-00412-t010:** PDI of unloaded NLCs before and after the stability test.

Formulations	Mean ± SD		
	Day 0	Day 30	Centrifuge
(2)	0.27 ± 0.01	0.28 ± 0.03	0.37 ± 0.04
(4)	0.26 ± 0.01	0.26 ± 0.02	0.26 ± 0.01
(5)	0.27 ± 0.01	0.27 ± 0.01	0.27 ± 0.01

**Table 11 pharmaceutics-16-00412-t011:** Zeta potential of unloaded NLCs before and after the stability test.

Formulations	Mean ± SD		
	Day 0	Day 30	Centrifuge
(2)	−54.90 ± 0.26	−63.70 ± 5.14 *	−58.33 ± 1.92
(4)	−53.27 ± 2.56	−72.97 ± 2.20 *	−57.30 ± 1.92
(5)	−53.57 ± 1.24	−84.97 ± 0.31 *	−58.30 ± 0.70

Asterisk (*) is significantly different at *p* < 0.05 when day 0 and day 30 are compared; centrifuge calculated by a Paired sample *t*-test using the SPSS program.

**Table 12 pharmaceutics-16-00412-t012:** Half-lives of capsaicin and curcumin in extract-loaded NLCs.

Compound	Half-Lives (h) Day 0	Half-Lives (h) Day 30
Capsaicin	38.50 ± 0.14	38.50 ± 0.14
Curcumin	57.75 ± 1.96	57.75 ± 1.96

## Data Availability

All data are contained within the article.

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
