# Peer review of "Natural Stabilizers and Nanostructured Lipid Carrier Entrapment for Photosensitive Compounds, Curcumin and Capsaicin"

_pharmaceutics, 2024, doi:10.3390/pharmaceutics16030412_

Round 1

Reviewer 1 Report

Comments and Suggestions for Authors

Natural Stabilizers and Nanostructured Lipid Carrier Entrapment for Photosensitive Compounds, Curcumin and Capsaicin

The authors address instability issues associated with chilli extract using anti oxidants and selected optimal levels of chili and turmeric extract ratio versus antioxidants. Thereafter, the select combination were encapsulated into a nanostructured lipid carrier. The formulation developed decreased the decomposition of active chemicals and enhanced bioavailability.

I would suggest that the following minor corrections are made:

·         Please add statistical significance values (p-values) to bar charts (e.g., Figure 2, 5 etc.)

Comments on the Quality of English Language

None

Author Response

Dear reviewer of Pharmaceutics,

We greatly appreciate the valuable comments and suggestions from all reviewers. We have carefully read and responded to all comments, point by point. The specific alterations in the manuscript in response to the reviewer comments are shown in yellow highlights for comments of reviewer 1. In addition, we have improved the figure quality following the academic editor comment. Others changes are shown in red letters.

We hope all of the changes have addressed the reviewers’ concern, so with these additions we hope our work will be accepted for publication in Pharmaceutics.

Best regards,

Asst. Prof. Dr. Kanokwan Kiattisin

Reviewer 2 Report

Comments and Suggestions for Authors

Authors conducted lots of tests on the stability, antioxidant, and encapsulation of two natural compounds. The overall study is interesting, but the data presentation must be improved. For all the half-life or IC50 studies, the authors presented them in the table as a number or a bar graph. There is no curve of original data points. Those must be added to the manuscript. Besides, there are too many tables of sizes and PDI. It will be better if authors can simplify them. Please find detailed comments below:

Check your numeric labels, many of them are disordered.

L67 What is the difference between three sunscreens? Are they particles or molecules?

Line 113 Clarify which chemical marker is used in each extract.

Table 1 What does Line ABC in the table mean?

L135-136 What are the active ingredients in tea and graph extract? Are they acting as sunscreen?

L 148& 150, what do [22] and [23] here mean?

L156-164, where are decay curves and data points?

L165-169, where are the data points showing this method presents the highest light stability?

Table 3: How did you determine the percentage of chemical markers in the extract?

Figure 1(i) and (j) What is the difference between chlorogenic acid standard and chlorogenic acid?

For all the assay data, how did you determine IC50?

L262-267, what is the difference between these two centrifugations?

L276, does NLC structure change during airdry? Have you used CryoEM to image them?

Figure 2 How many data points are in each group? Where are the original decay curves of those assays?

Figure 3 Where are the y-axis values?

Table 7 What is the reaction time of % Inhibition?

Line 600-602. Remove the last sentence from the conclusion. It’s a review of other people’s work and not related to the study in this paper.

Author Response

Dear reviewer of Pharmaceutics,

We greatly appreciate the valuable comments and suggestions from all reviewers. We have carefully read and responded to all comments, point by point. The specific alterations in the manuscript in response to the reviewer comments are shown in green highlights for the comments of reviewer 2. In addition, we have improved the figure quality following the academic editor comment. Others changes are shown in red letters.

We hope all of the changes have addressed the reviewers’ concern, so with these additions we hope our work will be accepted for publication in Pharmaceutics.

Best regards,

Asst. Prof. Dr. Kanokwan Kiattisin

Reviewer 3 Report

Comments and Suggestions for Authors

The manuscript deals with improving the photostability of capsaicin and curcumin with speculation of its application to pharmaceuticals. The work involved preparation and characterization of NLCs. However, the manuscript needs significant revisions to be accepted for publication in "Pharmaceutics". My specific comments are provided below:

Major comments:

1. The reporting is casual. Almost all the sections needs to concise. This includes - removal of unnecessary words and phrases, elimination of duplicacy of data (e.g. in table and text). 

2. The application to pharmaceuticals has not been addressed clearly. Authors need to carefully improve this part. 

3. Only the most relevant tables and figures should be kept in the MS. Other less important information should be moved to supplementary information. e.g. Figure 1.

4. The need of the problem, objectives and specific aims of the work should be clearly defined in the introduction, with mention of the intended pharmaceutical dosage form.  

5. The results should be interpreted in terms of improving the critical attributes of the targeted formulation.

Comments on the Quality of English Language

1. The reporting is casual. Almost all the sections needs to concise. This includes - removal of unnecessary words and phrases.

Author Response

Dear reviewer of Pharmaceutics,

We greatly appreciate the valuable comments and suggestions from all reviewers. We have carefully read and responded to all comments, point by point. The specific alterations in the manuscript in response to the reviewer comments are shown in purple highlights for the comments of reviewer 3. In addition, we have improved the figure quality following the academic editor comment. Others changes are shown in red letters.

We hope all of the changes have addressed the reviewers’ concern, so with these additions we hope our work will be accepted for publication in Pharmaceutics.

Best regards,

Asst. Prof. Dr. Kanokwan Kiattisin

Reviewer 4 Report

Comments and Suggestions for Authors

1. There is a disconnection between capsaicin and turmeric in the Abstract.
2. The main statement in the abstract from Line 15 to Line 17 is unclear.
3. There is no introduction or transition word for Chlorogenic acid in the abstract.
4. The abstract is not well-structured. Please work on a cohesive abstract by indicating the problems, solution, and result in short sentences.
5. Please add references to Line 38 and 39 to support the statement.
6. Please fix the grammar in Line 41.
7. Please add a short conclusion sentence for Line 46-48.
8. What is the focus of the application? In pharmaceutics or cosmetics?
9. The "Anathocyanin extracts" shown in the paragraph disconnected the whole introduction and distracted the focus of the manuscript.
10. The transition from sunscreen to NLC from Line 72 to 75 is not convincing. Is NLC better than sunscreen? Also, please add more references in Line 80.
11. The introduction is totally decentralized. The last paragraph in the introduction needs more information for how and what and then to guide the authors to determine if they are interested in the manuscript.
12. Please move the percent entrapment efficiency (EE%) to the methods.
13. Please organize the sentences in Line 145 to 150 more readable.
14. The whole manuscript is disconnected from Line 213 to Line 219.
15. In Figure 2, please remove the letters above each bar to improve readability.
16. The PDI in Table 9 is over 0.2 and even higher than 0.3. How to define a good PDI? Normally, 0.2 is optimal for lipid carriers, 0.3 is acceptable.
17. Conclusion is not a list of detailed results, it is a section to use data to support statements. Please work on developing an efficient conclusion

18. Please rethink about how to make a cohensive delivery of story telling manascript. 

Comments on the Quality of English Language

N/A

Author Response

Dear reviewer of Pharmaceutics,

We greatly appreciate the valuable comments and suggestions from all reviewers. We have carefully read and responded to all comments, point by point. The specific alterations in the manuscript in response to the reviewer comments are shown in blue highlights for the comments of reviewer 4. In addition, we have improved the figure quality following the academic editor comment. Others changes are shown in red letters.

We hope all of the changes have addressed the reviewers’ concern, so with these additions we hope our work will be accepted for publication in Pharmaceutics.

Best regards,

Asst. Prof. Dr. Kanokwan Kiattisin

Round 2

Reviewer 3 Report

Comments and Suggestions for Authors

1.        Please include the following information in the abstract

a.        Values of particle size and pdi

b.       Quantitative values of enhanced photostability rather than just writing 2 times

2.     The need of the problem, objectives and specific aims of the work should be clearly defined in the introduction, with mention of the intended pharmaceutical dosage form. This comment has not been carefully addressed.

3.     The results should be interpreted in terms of improving the critical attributes of the targeted formulation. This comment has not been carefully addressed. Please clearly define the critical attributes of the targeted formulation.

Comments on the Quality of English Language

Need minor editing.

Author Response

Dear Reviewer of Pharmaceutics,

We greatly appreciate the valuable comments and suggestions from all reviewers in Round 2. We have carefully read and responded to all comments, point by point. The specific alterations in the manuscript in response to the reviewer comments are shown in purple highlights for comments of reviewer 3. Others changes are shown in red letters.

We hope all of the changes have addressed the reviewers’ concern, so with these additions we hope our work will be accepted for publication in Pharmaceutics.

Best regards,

Asst. Prof. Dr. Kanokwan Kiattisin

Reviewer 4 Report

Comments and Suggestions for Authors

As a reviewer, I appreciate the great improvement that the authors have made. Please see the following adjustments:

- The current revised abstract has improved significantly, but it could be further enhanced by briefly stating the hypothesis clearly. The reason for using NLCs is not indicated in the abstract. Additionally, detailed information about NLCs is not necessary and should be moved to the introduction.

- The response regarding "Anthocyanin extracts" does not contribute to the manuscript's improvement. If the authors believe that including this paragraph can help explain the mechanisms of photostability, they should integrate it into the sentences rather than introducing anthocyanin extracts abruptly.

- There is a grammar issue in the first sentence of page 2, lines 93-99.

- For lines 145-150, I suggest the authors consider using a table.

- I could not understand the sentences on page 14, lines 520-521. Therefore, I disagree with the statement regarding the "Narrow PDI value" in the conclusion.

- Please add quantitative descriptions to the conclusion paragraphs.

Comments on the Quality of English Language

n/a

Author Response

Dear Reviewer of Pharmaceutics,

We greatly appreciate the valuable comments and suggestions from all reviewers in Round 2. We have carefully read and responded to all comments, point by point. The specific alterations in the manuscript in response to the reviewer comments are shown in blue highlights for comments of reviewer 4. Others changes are shown in red letters.

We hope all of the changes have addressed the reviewers’ concern, so with these additions we hope our work will be accepted for publication in Pharmaceutics.

Best regards,

Asst. Prof. Dr. Kanokwan Kiattisin
